# A Digital-Simulation Model for a Full-Polarized Microwave Radiometer System and Its Calibration

Jia Ding [1,2], Zhenzhan Wang [1,*], Yongqiang Duan [1,2], Xiaolin Tong [1] and Hao Lu [1]

1   Key Laboratory of Microwave Remote Sensing, National Space Science Center, Chinese Academy of Sciences, Beijing 100190, China; dingjia19@mails.ucas.ac.cn (J.D.); duanyongqiang15@mails.ucas.ac.cn or eece@ucas.ac.cn (Y.D.); tongxiaolin@nssc.ac.cn (X.T.); luhao@mirslab.cn (H.L.)
2   School of Electronic, Electrical and Communication Engineering, University of Chinese Academy of Sciences, Beijing 100049, China
*   Correspondence: wangzhenzhan@mirslab.cn; Tel.: +86-10-6258-6454

**Abstract:** A digital-correlation full-polarized microwave radiometer is an important passive remote sensor, as it can obtain the amplitude and phase information of an electromagnetic wave at the same time. It is widely used in the measurement of sea surface wind speed and direction. Its configuration is complicated, so the error analysis of the instrument is often difficult. This paper presents a full-polarized radiometer system model that can be used to analyze various errors, which include input signal models and a full-polarized radiometer (receiver) model. The input signal models are generated by WGN (white Gaussian noise), and the full-polarized radiometer model consists of an RF front-end model and digital back-end model. The calibration matrix is obtained by solving the overdetermined equations, and the output voltage is converted into Stokes brightness temperature through the calibration matrix. Then, we use the four Stokes parameters to analyze the sensitivity, linearity, and calibration residuals, from which the simulation model is validated. Finally, two examples of error analysis, including gain imbalance and quantization error, are given through a simulation model. In general, the simulation model proposed in this paper has good accuracy and can play an important role in the error analysis and pre-development of the fully polarized radiometer.

**Keywords:** digital-correlation full-polarized radiometer; modeling of microwave system; full-polarized calibration target

## 1. Introduction

A full-polarized microwave radiometer is a type of passive microwave remote sensor developed in the mid-1990s, represented by the WindSat spaceborne polarimetric microwave radiometer [1,2] and the Water Cycle Observation Mission (WCOM) in China [3,4]. It is based on the traditional microwave radiometer to measure the first two Stokes parameters of the target radiation, further increasing the ability to measure the third and fourth Stokes parameters, and realizing the simultaneous measurement of the target microwave radiation amplitude and phase [5]. The digital-correlation full-polarized microwave radiometer uses a digital correlator to implement a complex multiplication of the vertically and horizontally polarized components of the electric field, extracting the real and imaginary parts of the product to obtain the third and fourth Stokes parameters, which contain information about the sea surface wind field.

Calibration of a radiometer is a prerequisite for quantitative remote sensing. In order to calibrate a full-polarized radiometer, four Stokes parameters with known brightness temperature must be generated from a calibration target. There are two calibration methods, the first one is to generate four known Stokes parameters by a programmable active noise source [6]. However, this method does not include the effects of the antenna and is therefore

not an end-to-end calibration. The second one uses a combination of a hot and a cold blackbody by a freestanding metal grid to produce the first-three linearly-polarized Stokes brightness temperatures [7], followed by a parallel-grooved dielectric plate to generate a circularly polarized wave [8,9]. Currently, NOAA and HUT have designed full-polarized passive calibration targets (FPCS), according to the theory above [10,11]. In addition, based on FPCS, the National Space Science Center, Chinese Academy of Sciences (NSSC, CAS) also designed a compact polarimeter calibration target (CPCT) with an improvement on phase plate and grid rotation mode, which needs a simpler operation during calibration [12].

The full-polarized radiometer and its calibration process are more complicated than traditional radiometers, so many unknown errors may be introduced in the calculation of all Stokes parameters, such as the effects of gain imbalance, channel crosstalk, and phase error, which are difficult to analyze by theory or experiment. By accurately modeling a full-polarized radiometer and using the simulation model we can simplify the error analysis for a full-polarized radiometer. In addition, the simulation model is very important for the development of a new full-polarized radiometer; when the real one works, a digital model also provides great convenience for data analysis and fault monitoring. Many researchers have worked on the modeling and simulation of traditional radiometers. A simulation model for a total-power radiometer was established through simulink, which only gave some qualitative simulation results [13]. A modeling method for comprehensive simulation of radiometer channel response is proposed in [14], this method is based on quashing universal circuit simulator (QUCS), which gives the flexibility of choosing among the available devices. A pseudo-random number was used to simulate the waveform of the input noise signals to the integrator output signals (before A/D conversion), and the linearity of the simulation model was validated [15]. In order to estimate the sensitivity more accurately, a video voltage signal model based on the front-end and back-end amplitude spectral density (ASD) was proposed for a 52 GHz total power radiometer [16]. A system simulation model based on digital signal processing was presented for a total power radiometer, which was used to estimate the output dynamic range, linearity, and sensitivity [17]. A simulation model for a spectrometer was proposed in [18]. Although there has been research on the modeling of a total-power radiometer, there is currently little related work on modeling full-polarized radiometers. This has caused problems for the research into a full-polarized radiometer.

Based on the CPCT system [19], this paper presents an end-to-end simulation method of a full-polarized radiometer system. The model includes input signal models and a full-polarized radiometer model. The input signals are used as input to the full-polarized radiometer model to generate output voltages of four Stokes parameters, which include hot and cold calibration target signals, ideal full-polarized signals, as well as CPCT signals. The output brightness temperature of the CPCT system (the input brightness temperature of the receiver) and the output voltage of the receiver can be used for calibration. Through calibration, the four Stokes parameters of the full-polarized signal can be determined. These four Stokes brightness temperature parameters, especially the latter two, are of great significance for the measurement of sea surface wind field. By calculating the sensitivity, linearity, and calibration accuracy of the system, the simulation model is validated. In addition, we applied the radiometer model to the error analysis of gain imbalance and quantization, and gave their effects on results. Section 2 introduces the digital-correlation full-polarized radiometer system and its calibration principle. Section 3 introduces the modeling of the full-polarized radiometer system. Section 4 gives simulation results and calculates some indicators. It also presents an error analysis for gain imbalance and quantization. Section 5 summarizes this paper.

## 2. Full-Polarized Radiometer and Its Calibration

### 2.1. Digital-Correlation Full-Polarized Radiometer

The radiation field generated by the target will pass through the atmosphere, finally be received by the antenna, and propagate in the receiver in the form of voltage. Due to

space limitations, this paper mainly introduces the internal model of the radiometer. The digital-correlation full-polarized radiometer uses a digital correlator to simultaneously realize the autocorrelation and cross-correlation of vertically and horizontally polarized signals. The block diagram of the radiometer is shown in Figure 1.

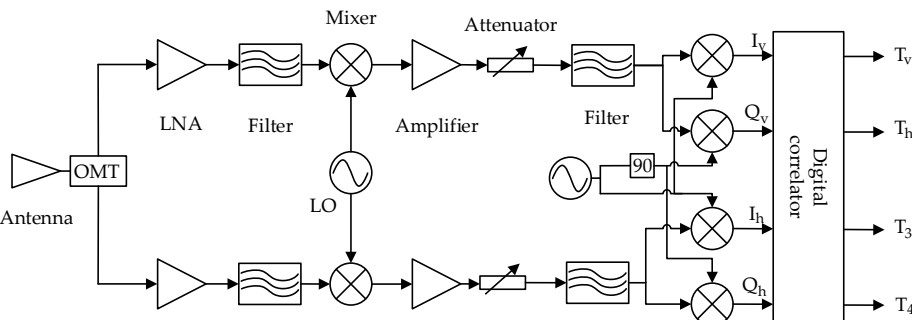

**Figure 1.** Process of the proposed MRID.

It can be seen that the radiometer is mainly composed of a dual-polarized antenna, a superheterodyne receiver with IQ detection (front-end), and a digital correlator. The antenna receives the target's vertically and horizontally polarized signals. These signals are separated by the orthomode transducer (OMT) and input into the front-end. The IQ signals output from the front-end are correlated in the digital correlator. Currently, digital correlators are widely used in full-polarized radiometers. Therefore, a digital-correlation full-polarized radiometer is modeled in this paper. Furthermore, in this paper, we use a direct correlation correlator.

According to the definition of the modified Stokes parameters, the digital correlator uses the following four equations to calculate the output voltages of four Stokes parameters [20]:

$$
\begin{aligned}
V_v &=< I_v(n) \times I_v(n) + Q_v(n) \times Q_v(n) >, \\
V_h &=< I_h(n) \times I_h(n) + Q_h(n) \times Q_h(n) >, \\
V_3 &= < I_v(n) \times I_h(n) + Q_h(n) \times Q_v(n) >, \\
V_4 &=< I_h(n) \times Q_v(n) - I_v(n) \times Q_h(n) >;
\end{aligned}
\tag{1}
$$

where $I_v = S_v S_I$, $Q_v = S_v S_Q$, $I_h = S_h S_I$, and $Q_h = S_h S_Q$. $S_I$ and $S_Q$ are two orthogonal modulated signals with a phase difference of $\pi/2$. $I_v$ and $Q_v$ represent orthogonal signals of vertically polarized signals, and $I_h$ and $Q_h$ represent those of horizontally polarized signals. $V_v$ and $V_h$ represent voltages of the vertically and horizontally polarized signals, respectively; $V_3$ and $V_4$ represent voltages of the third and fourth Stokes parameters, respectively; and n is the number of signal points. By averaging multiple sets of output signals, the output voltages with an expected integration time can be obtained. The simulation model in this paper is based on a 19.35 GHz digital-correlation full-polarized radiometer designed by NSSC. The main characteristics of the radiometer are shown in Table 1.

**Table 1.** The main characteristics of 19.35 GHz full-polarized radiometer.

| LO Frequency | Bandwidth | System Noise | Integration Time | Receiver Typ |
|---|---|---|---|---|
| 18.75 GHz | 750 MHz | 300 K | 3 ms | SSB |

## 2.2. Compact Passive Microwave Calibration Target (CPCT)

The CPCT, proposed by NSSC, inherits the traditional full-polarized calibration target in principle, which uses two blackbodies of different brightness temperatures and a

polarization grid to generate stable linearly polarized signals. Then, when the signals pass through the wave plate, which acts as a phase retarder, the linearly polarized signals are changed into circularly polarized signals. When the angle of the polarization grid or the distance of the phase retarder is changed, the output of the system also changes. The principle of CPCT is shown in Figure 2.

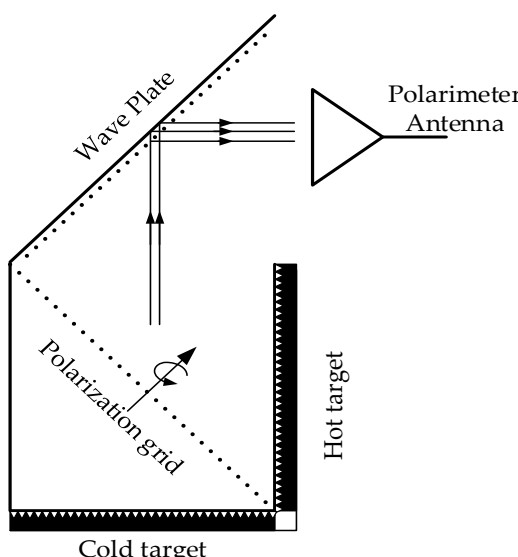

**Figure 2.** Principle of CPCT.

Since the parameters of various materials and components of the calibration target are known, the output brightness temperature signals can be calculated, and controllable full-polarized microwave radiation noise signals are generated, which can be performed for end-to-end calibration of the full-polarized radiometer.

According to [6], the first-three Stokes parameters after the polarization grid can be written as:

$$
\begin{bmatrix} T_v' \\ T_h' \\ T_3' \end{bmatrix} = \begin{bmatrix} cos^2(\theta) & sin^2(\theta) \\ sin^2(\theta) & cos^2(\theta) \\ -sin(2\theta) & sin(2\theta) \end{bmatrix} \begin{bmatrix} r_{v1}^2 & 0 \\ 0 & r_{p1}^2 \end{bmatrix} \begin{bmatrix} T_H \\ T_H \end{bmatrix}
$$
$$
+ \begin{bmatrix} cos^2(\theta) & sin^2(\theta) \\ sin^2(\theta) & cos^2(\theta) \\ sin(2\theta) & -sin(2\theta) \end{bmatrix} \begin{bmatrix} t_{v1}^2 & 0 \\ 0 & t_{p1}^2 \end{bmatrix} \begin{bmatrix} T_C \\ T_C \end{bmatrix},
\tag{2}
$$

where $T_H$ and $T_C$ are brightness temperature of the hot and cold target, respectively; $T_v'$, $T_h'$, and $T_3'$ are the first three Stokes brightness temperatures; $r_{p1}$ and $t_{p1}$ are reflection coefficient and transmission coefficient for the polarized signals parallel to the polarization grid wires, respectively; the parameters for the polarized signals perpendicular to the grid wires are $r_{v1}$ and $t_{v1}$; and $\theta$ is the polarization grid orientation angle measured with respect to the antenna polarization basis. Then, the four output Stokes parameters of the CPCT can be written as [19]:

$$
\begin{aligned}
T_v &= T_v'\left(r_{v2}^2 + t_{v2}^4 + 2r_{v2}t_{v2}^2 cos\xi\right), \\
T_h &= T_h'\left(r_{p2}^2 + t_{p2}^4 + 2r_{p2}t_{p2}^2 cos\xi\right), \\
T_3 &= T_3'\left[(r_{p2}r_{v2}) + \left(t_{p2}^2 t_{v2}^2\right) + \left(r_{v2}t_{p2}^2 + r_{p2}t_{v2}^2\right)cos\xi\right], \\
T_4 &= T_3'\left(-r_{v2}t_{p2}^2 + r_{p2}t_{v2}^2\right)sin\xi;
\end{aligned}
\tag{3}
$$

where $T_v$, $T_h$, $T_3$, and $T_4$ are the output Stokes brightness temperatures from the CPCT; the subscripts '2' denote the reflectivity $r$ and transmittance $t$ of the wave plate's grid, which are different to those of polarization grid; and $\zeta$ is the phase difference produced by the wave plate. Since all the parameters of the various materials and components of the calibration target are known, the output brightness temperature of the CPCT system can be calculated from Equations (2) and (3).

The output brightness temperatures of the CPCT system are the input brightness temperatures of the full-polarized radiometer. The relationship between the output voltages and the input brightness temperatures of the full-polarized radiometer can be expressed as [10]:

$$
\bar{V} = \begin{bmatrix} V_v \\ V_h \\ V_3 \\ V_4 \end{bmatrix} = \begin{bmatrix} g_{vv} & g_{vh} & g_{v3} & g_{v4} \\ g_{hv} & g_{hh} & g_{h3} & g_{h4} \\ g_{3v} & g_{3h} & g_{33} & g_{34} \\ g_{4v} & g_{4h} & g_{43} & g_{44} \end{bmatrix} \begin{bmatrix} T_v \\ T_h \\ T_3 \\ T_4 \end{bmatrix} + \begin{bmatrix} o_v \\ o_h \\ o_3 \\ o_4 \end{bmatrix}, \tag{4}
$$

where $\bar{V}$ is the output voltage; $\bar{\bar{g}}$ and $\bar{o}$ are the gain matrix and offset matrix of the full-polarized radiometer, respectively. In the gain matrix, elements other than the diagonal represent crosstalk between channels. In order to solve the matrix conveniently, we rewrite Equation (4) in the following form:

$$
\bar{V} = \begin{bmatrix} V_v \\ V_h \\ V_3 \\ V_4 \\ 1 \end{bmatrix} = \begin{bmatrix} g_{vv} & g_{vh} & g_{v3} & g_{v4} & o_v \\ g_{hv} & g_{hh} & g_{h3} & g_{h4} & o_h \\ g_{3v} & g_{3h} & g_{33} & g_{34} & o_3 \\ g_{4v} & g_{4h} & g_{43} & g_{44} & o_4 \\ 0 & 0 & 0 & 0 & 1 \end{bmatrix} \begin{bmatrix} T_v \\ T_h \\ T_3 \\ T_4 \\ 1 \end{bmatrix}, \tag{5}
$$

when there is more than one set of inputs and outputs, $\bar{V}$ was rewritten as $\bar{\bar{V}}$, and $\bar{T}$ was rewritten as $\bar{\bar{T}}$. Therefore, the calibration matrix:

$$
\bar{\bar{G}} = \bar{V} \left( \bar{\bar{T}} \right)^{-1}. \tag{6}
$$

During the calibration process, the radiometer observes the independent brightness temperature under different conditions; $\bar{\bar{G}}$ can be obtained through at least five sets of output voltages and input brightness temperatures. This requires the five sets of data to be linearly independent. In order to obtain a more accurate calibration matrix, more than five groups of data are often used to achieve the calibration, which are divided into several categories shown in Table 2. Then the calibration matrix can be written as [12]:

$$
\bar{\bar{G}} = \bar{V} \left( \bar{\bar{T}} \right)^T \left[ \bar{\bar{T}} \left( \bar{\bar{T}} \right)^T \right]^{-1}. \tag{7}
$$

**Table 2.** Data types used to calculate the calibration matrix.

| Target Type | Hot Target | Cold Target | Grid Rotation Angle | Delay Phase |
|---|---|---|---|---|
| Fully-polarized target | 300 K | 80 K | 0:0.04π:2π | −π/4 |
| | 300 K | 80 K | 0:0.04π:2π | −7π/4 |
| Non-polarized target | 300 K | / | Without CPCT system | |
| | / | 80 K | Without CPCT system | |

## 3. Simulation Model

The full-polarized radiometer system model is composed of input signal models and a full-polarized radiometer model. The input signal models generate full-polarized signals, which are used as the inputs of the full-polarized radiometer model.

### 3.1. Input Signal Models

In order to evaluate and validate the performance of this system model, input signals need to be generated firstly, including hot and cold target signals and CPCT signals which are both used for calibration; ideal full-polarized signals are used to simulate the target scene temperatures.

### 3.1.1. Hot and Cold Calibration Target Signals

Hot and cold targets are generally made from blackbody and are non-polarized, so $T_v$ and $T_h$ are equal, and $T_3$ and $T_4$ are zero. Taking a hot target as an example, the vertically and horizontally polarized signals of hot target can be expressed as:

$$S_{hot,v}(n) = wgn\left(\sigma_{hot,v}^2\right),$$
$$S_{hot,h}(n) = wgn\left(\sigma_{hot,h}^2\right), \tag{8}$$

where $S_{hot,v}(n)$ and $S_{hot,h}(n)$ are the vertically and horizontally polarized signals, respectively; wgn represents a function that can generate white Gaussian noise( WGN) with a mean of 0 and a variance of $\sigma^2$. Because $S_{hot,v}(n)$ and $S_{hot,h}(n)$ are generated independently and are not correlated, the corresponding third and fourth Stokes temperatures $T_3$ and $T_4$ should be zero. The variance of WGN for the hot target and cold target can be expressed as:

$$\sigma_{hot,v}^2 = kT_{hot,v}B,$$
$$\sigma_{hot,h}^2 = kT_{hot,h}B, \tag{9}$$

where $T_{hot,v}$ and $T_{hot,h}$ represent the vertically and horizontally polarized brightness temperatures of hot target, respectively. They are all equal to the brightness temperature of the hot target, so $\sigma_{hot,v}^2 = \sigma_{hot,h}^2 = \sigma_{hot}^2$, $k$ is the Boltzmann constant, and $B$ is the bandwidth.

### 3.1.2. CPCT Signals

CPCT is capable of generating four Stokes parameters with known brightness temperatures. It can be seen from Figure 2 that CPCT includes a hot target and a cold target. Therefore, firstly, the output signals of the hot and cold calibration targets are generated by:

$$S_{hot,v}(n) = wgn_1\left(\sigma_{hot}^2\right),$$
$$S_{hot,h}(n) = wgn_2\left(\sigma_{hot}^2\right),$$
$$S_{cold,v}(n) = wgn_1\left(\sigma_{cold}^2\right),$$
$$S_{cold,h}(n) = wgn_2\left(\sigma_{cold}^2\right); \tag{10}$$

where $\sigma_{hot}^2$ and $\sigma_{cold}^2$ are the variance of hot target and cold target signals, respectively; the subscripts '1' and '2' mean two signals are generated independently. The signals $\left[S_{g,v}, S_{g,h}\right]'$ after the polarization grid are [19]:

$$\begin{bmatrix} S_{g,v} \\ S_{g,h} \end{bmatrix} =$$
$$\begin{bmatrix} cos\alpha_1 & sin\alpha_1 \\ -sin\alpha_1 & cos\alpha_1 \end{bmatrix} \begin{bmatrix} r_{v1} & 0 \\ 0 & r_{p1} \end{bmatrix} \begin{bmatrix} cos\alpha_1 & -sin\alpha_1 \\ sin\alpha_1 & cos\alpha_1 \end{bmatrix} \begin{bmatrix} S_{hot,v} \\ S_{hot,h} \end{bmatrix} +$$
$$\begin{bmatrix} cos\beta_1 & -sin\beta_1 \\ sin\beta_1 & cos\beta_1 \end{bmatrix} \begin{bmatrix} t_{v1} & 0 \\ 0 & t_{p1} \end{bmatrix} \begin{bmatrix} cos\beta_1 & sin\beta_1 \\ -sin\beta_1 & cos\beta_1 \end{bmatrix} \begin{bmatrix} S_{cold,v} \\ S_{cold,h} \end{bmatrix}. \tag{11}$$

The signals $[S_{d,v}, S_{d,h}]'$ passing through the wave plate's grid are:

$$
\begin{bmatrix} S_{d,v} \\ S_{d,h} \end{bmatrix} = \begin{bmatrix} \cos\alpha_2 & \sin\alpha_2 \\ -\sin\alpha_2 & \cos\alpha_2 \end{bmatrix} \begin{bmatrix} t_{v2} & 0 \\ 0 & t_{p2} \end{bmatrix} \begin{bmatrix} \cos\alpha_2) & -\sin\alpha_2 \\ \sin\alpha_2) & \cos\alpha_2 \end{bmatrix}
$$
$$
\begin{bmatrix} \cos\beta_2 & -\sin\beta_2 \\ \sin\beta_2 & \cos\beta_2 \end{bmatrix} \begin{bmatrix} t_{v2} & 0 \\ 0 & t_{p2} \end{bmatrix} \begin{bmatrix} \cos\beta_2 & \sin\beta_2 \\ -\sin\beta_2 & \cos\beta_2 \end{bmatrix} \begin{bmatrix} S_{g,v}e^{j\xi} \\ S_{g,h}e^{j\xi} \end{bmatrix} +
$$
$$
\begin{bmatrix} \cos\beta_2 & -\sin\beta_2 \\ \sin\beta_2 & \cos\beta_2 \end{bmatrix} \begin{bmatrix} r_{v2} & 0 \\ 0 & r_{p2} \end{bmatrix} \begin{bmatrix} \cos\beta_2 & \sin\beta_2 \\ -\sin\beta_2 & \cos\beta_2 \end{bmatrix} \begin{bmatrix} S_{g,v} \\ S_{g,h} \end{bmatrix}. \tag{12}
$$

The two signals are the vertically and horizontally polarized input voltages of the full-polarized radiometer. $\alpha_1$ and $\beta_1$ are the projections of the polarization gird rotation angle in the vertical plane and the horizontal plane, respectively. Since the grid plane angle is $45°$, $\alpha_1$ and $\beta_1$ are equal ($\theta$). Furthermore, so is the relationship between $\alpha_2$, $\beta_2$, and the wave plate's grid.

### 3.1.3. Ideal Full-Polarized Signals

Ideal full-polarized signals can be represented by a vertically polarized signal $S_{i,v}(n)$, a horizontally polarized signal $S_{i,h}(n)$, and the phase difference $\phi$ between them. If two noise signals are generated independently, they are uncorrelated, and $T_3$ and $T_4$ would be zero. Therefore, we need to generate two signals with certain correlation. Assuming the brightness temperature of vertically polarized signal is $T_v$, then, it can be generated by Equation (8). In order to generate a horizontally polarized signal with a brightness temperature of $T_h$ and a phase difference of $\phi$, we need to multiply $S_{i,v}(n)$ by the phase delay coefficient $e^{-j\phi}$. According to the property of the WGN, the amplitude of $S_{i,h}(n)$ is $\sqrt{T_h/T_v}$ times that of $S_{i,v}(n)$. Therefore, the signal $S_{i,h}(n)$ can be expressed as:

$$
S_{i,h}(n) = \sqrt{T_h/T_v} S_{i,v}(n)e^{-j\phi}, \tag{13}
$$

where $e^{-j\phi}$ means that the phase of horizontally polarized signal is $\phi$ behind that of vertically polarized signal.

### 3.2. Full-Polarized Radiometer Mode
### 3.2.1. RF Front-End Model

The RF front-end includes vertically and horizontally polarized channels. The two channels must be identical in design. Therefore, the gain, system noise, and bandwidth of the two channels in the model are all set to the same. Noise in the RF front-end can be expressed in terms of equivalent noise temperature. Assuming that the radiometer receiver noise of both channels is $T_{rec}$, which are not correlated with each other, then the receiver noise signals are generated by:

$$
S_{rec,v}(n) = wgn_1\left(\sigma_{rec}^2\right),
$$
$$
S_{rec,h}(n) = wgn_2\left(\sigma_{rec}^2\right), \tag{14}
$$

where $S_{rec,v}(n)$ and $S_{rec,h}(n)$ are the noise signals of vertically and horizontally polarized channels, respectively. Assuming the gain of both channels is A, and the Butterworth filter is used, then, the vertically and horizontally polarized signals after being amplified and filtered can be expressed as [21]:

$$
S_{f,v}(n) = A \cdot H_{Butter}[S_{rec,v}(n) + S_{d/i,v}(n)],
$$
$$
S_{f,h}(n) = A \cdot H_{Butter}[S_{rec,h}(n) + S_{d/i,h}(n)], \tag{15}
$$

where $H_{Butter}(\cdot)$ represents the transfer function of Butterworth filter; $S_{d/i,v}(n)$ and $S_{d/i,h}(n)$ are vertically and horizontally polarized signals above. IQ demodulation generates IQ

signals with a phase difference of 90° by mixing two LO signals with a phase difference of 90° with the input signals. According to the working principle of the digital-correlator, the output IQ signals are [22]:

$$
\begin{aligned}
I_v(n) &= S_{f,v}(n) \cdot S_{I,v}(n), \\
Q_v(n) &= S_{f,v}(n) \cdot S_{Q,v}(n), \\
I_h(n) &= S_{f,h}(n) \cdot S_{I,h}(n), \\
Q_h(n) &= S_{f,h}(n) \cdot S_{Q,h}(n).
\end{aligned}
\tag{16}
$$

The IQ signals are the outputs of RF front-end model and are used as the inputs of digital correlator model.

### 3.2.2. Digital Correlator Model

The digital correlator contains quantization, multiple correlation, and integration of the IQ signals. Firstly, the IQ signals are quantized. Taking $I_v(n)$ as an example, quantization can be expressed as [23]:

$$
\hat{I}_v(n) = Q_i[I_v(n)]
\tag{17}
$$

where $\hat{I}_v(n)$ is quantized signal; $Q_i[\cdot]$ represents $i$-bit quantization. The voltages of four Stokes parameters before integration are obtained by autocorrelation and cross-correlation of the quantized IQ signals by Equation (1). Assuming that the integration time is $\tau ms$, and the sampling time of the simulation system is $t_s ms$, which is equivalent to performing $\tau/t_s$ accumulation operations on the above signals, finally, the output voltages of the digital correlator at a specific integration time can be obtained.

## 4. Results

In this section, the results of the output signals are given firstly and then the full-polarized radiometer system model is validated by calculating and analyzing three indicators of the radiometer, including the sensitivity, linearity, and calibration residual error.

### 4.1. Simulation Results

### 4.1.1. Output Signals

In order to speed up the simulation, we can ignore the unimportant bandwidth and focus only on the part slightly larger than the IF bandwidth (750 MHz), so the simulation bandwidth is set to 1 GHz (18.75 GHz − 19.75 GHz), and the sampling frequency is 2 GHz. $rp_1 = 0.998$, $rv_1 = 0.002$, $tv_1 = 0.998$, $tp_1 = 0.002$; $rp_2 = 0.997$, $rv_2 = 0.003$, $tv_2 = 0.997$, and $tp_2 = 0.003$. The $T_v$ and $T_h$ of ideal full-polarized signals are both set to 200 K, and the phase difference is set to 45°, then, the ideal $T_3$ and $T_4$ should be 282.8 K and −282.8 K. The brightness temperature of the hot target and cold target are set to 300 K and 80 K, respectively. In the simulation model, other settings are consistent with Table 1. The target signals are amplified and mixed by the front-end model through Equation (15). The vertically and horizontally polarized signal spectrums of ideal fully polarized signals are shown in Figure 3.

As can be seen from Figure 3, because $T_v$ and $T_h$ are equal, the spectrum of the vertically polarized and horizontally polarized signals is basically the same. After the signals are mixed and amplified, the IF bandwidth is 750 MHz.

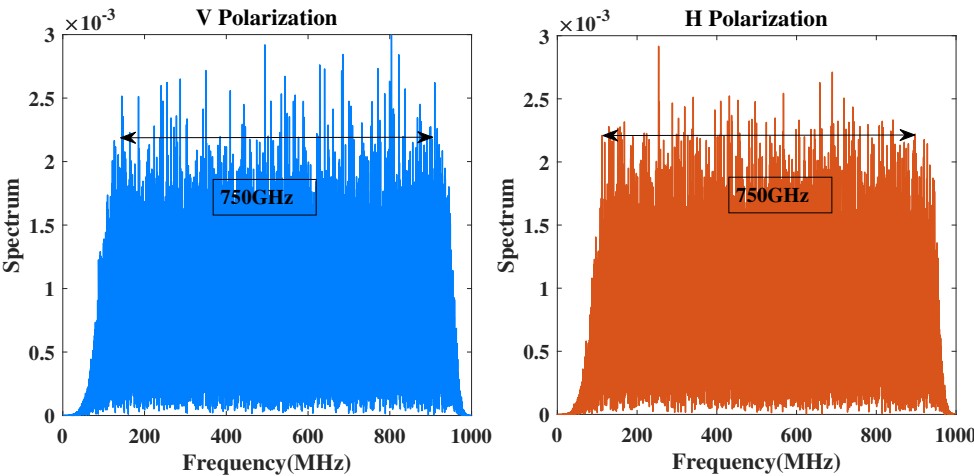

**Figure 3.** Spectrum of vertical and horizontal channel signals after mixer.

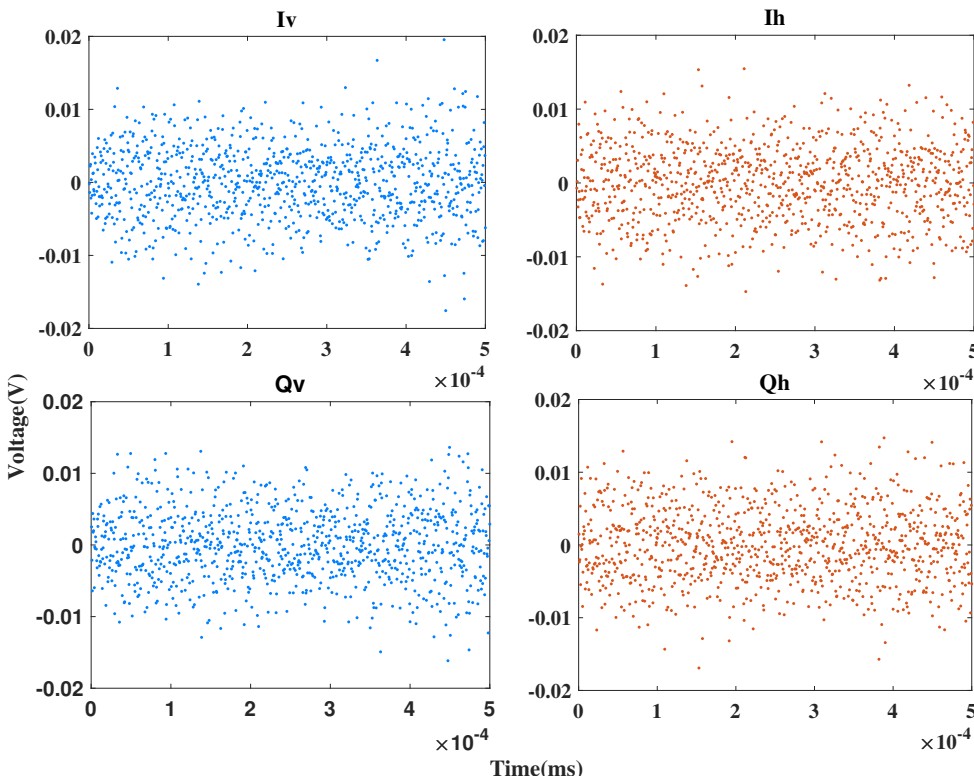

**Figure 4.** IQ signals of IQ demodulation.

Figure 4 shows the IQ signals after IQ demodulation through Equation (16). The phase difference between two signals can be obtained by the autocorrelation and cross-correlation of two signals, which is shown as:

$$
\begin{aligned}
r(x,y) &= \frac{Cov(x,y)}{\sqrt{Var(x)Var(y)}}, \\
\varphi(x,y) &= arctan\left(\frac{Im[r(x,y)]}{Re[r(x,y)]}\right),
\end{aligned}
\tag{18}
$$

where $r(x,y)$ represents the cross-correlation coefficient; $\varphi(x,y)$ is the phase difference between $x$ and $y$ . After calculation, the phase difference of full-polarized signals $\varphi(I_v, Q_v)$ is $-89.99°$, $\varphi(I_v, I_h)$ is $45.60°$, and $\varphi(I_v, Q_h)$ is $-44.40°$, which are consistent with the theoretical values.



The digital correlator model is used to perform autocorrelation and cross-correlation through Equation (1). The output voltages of the hot and cold target signals, CPCT signals, and the ideal full-polarized signals are shown in Figure 5, where the receiver noise is 300 K, and the integration time is 0.03 ms.

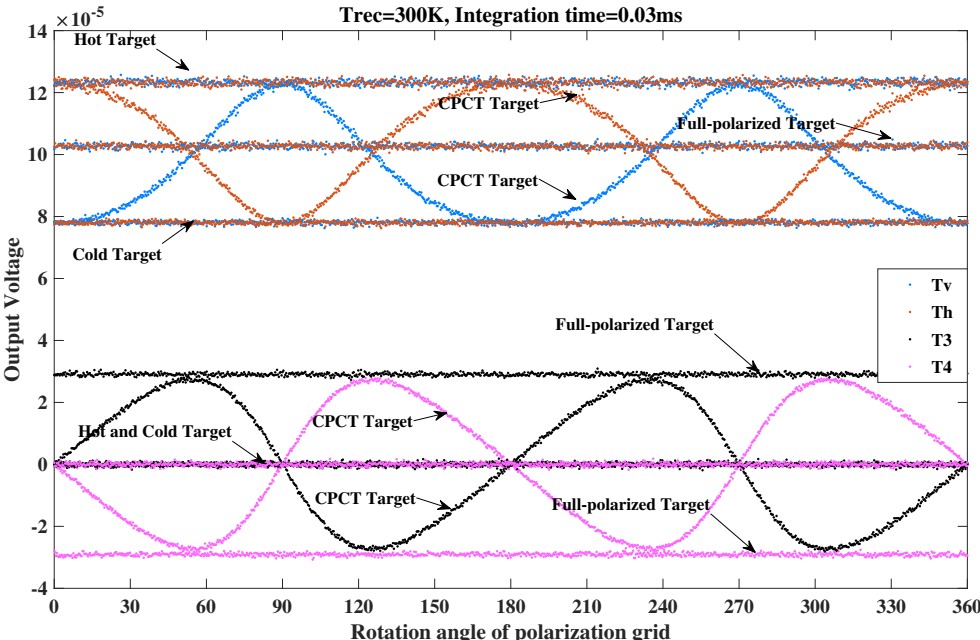

**Figure 5.** Output four Stokes voltages of different targets.

Firstly, it can be seen that $V_v$ and $V_h$ of the hot target and cold target are equal, but $V_3$ and $V_4$ are both zero, which is consistent with the theory that the hot target and cold target are nonpolarized. Secondly, it can be seen that the CPCT generates four Stokes voltages that change with the polarization grid angle. The maximum and minimum of the $V_3$ and $V_4$ of the CPCT are close to the $V_v$ and $V_h$ of the hot target and cold target, and $V_3$ and $V_4$ are symmetrical about zero, which is consistent with Equation (3). Thirdly, the $V_v$ and $V_h$ of the ideal full-polarized signals are equal, and $V_3$ and $V_4$ are symmetrical about zero. This is because the $T_v$ and $T_h$ of the ideal full-polarized signals are both set to 200 K, and their phase difference is $45°$. After the above output voltages are obtained, the calibration matrix is calculated from the voltages and brightness temperatures of the CPCT at different angles and the hot and cold target. Finally, the brightness temperature of the ideal fully polarized signals can be determined by a calibration matrix $\bar{\bar{G}}^{-1}$.

In fact, the radiation on the sea surface is not an ideal fully polarized wave, and also the coherence angle of horizontal and vertical polarization is not a fixed value. When the coherence angle is fixed at $45°$, and the ratio of the polarization part to the whole is set to 1/10, 1/5, 1/2, and 1/1, respectively, the theoretical brightness temperature and calculated value are shown in Figure 6. When the coherence angle of the ideal fully polarized wave is set to $0°$, $30°$, $45°$, $60°$, $90°$, respectively, the theoretical brightness temperature and calculated value are shown in Figure 7.

It can be seen from Figures 6 and 7 that the simulation results are strongly consistent with the theoretical values, and the error can be controlled within 0.7%. When the integration time increases, this error will become smaller. The third and fourth Stokes parameters of the actual sea surface are very small, so a long integration time is necessary. It shows that the model has the ability to simulate sea surface radiation. However, in order to facilitate the calculation, in subsequent experiments the ideal full polarization signal is still used to analyze the model.

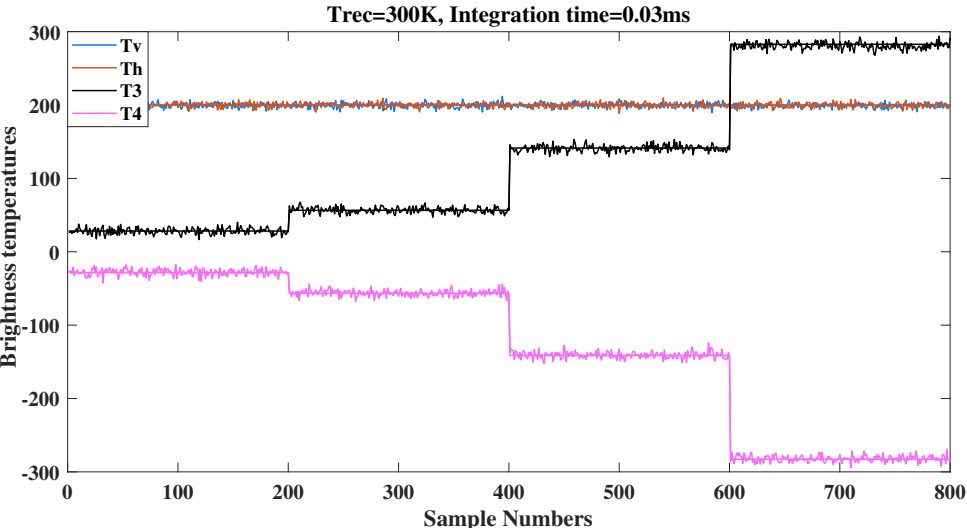

**Figure 6.** Output four Stokes voltages of different targets.

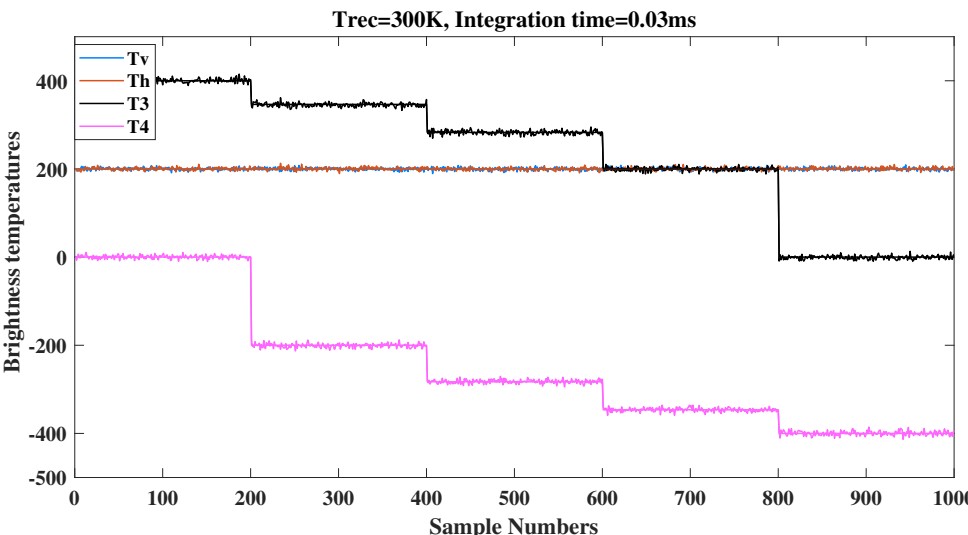

**Figure 7.** Output four Stokes voltages of different targets.

4.1.2. Calibration result

The corresponding relationship between Stokes brightness temperatures and output voltages is $\bar{\bar{T}} = \bar{\bar{G}}^{-1}\bar{V}$; when the receiver noise is 300 K and integration time is 0.03 ms, the results are as follows.

$$
\begin{bmatrix}
4861179 & 596.5205 & 2183.123 & -3321.299 & -299.3865 \\
3081.470 & 4863922 & 3780.104 & 4419.310 & -299.8512 \\
3784.458 & -2210.244 & 9726745 & -3360.274 & -0.069118 \\
1316.077 & -2323.335 & 1801.983 & 9727313 & 0.098348 \\
-1.2E^{-11} & -2.6E^{-12} & 1.8E^{-14} & 5.0E^{-14} & 1.000000
\end{bmatrix}
$$

$\bar{\bar{G}}^{-1}(1,5)$ and $\bar{\bar{G}}^{-1}(2,5)$ represent the vertically and horizontally polarized brightness temperature of receiver noise, which are consistent with the theoretical values.

Through $\bar{\bar{G}}^{-1}$, the output Stokes brightness temperatures of CPCT can be calibrated, and the results are shown in Figure 8a. It can be seen that $T_3$ and $T_4$ are symmetrical about zero, and the Stokes brightness temperatures change with rotation angle of the grid, respectively; which are basically consistent with the theoretical values. The four Stokes brightness temperatures of the ideal full-polarized signals can be calculated, and the results

are shown in Figure 8b. The means of the four Stokes parameters are 200.11 K, 199.95 K, 282.86 K, and −282.42 K, which are consistent with the input brightness temperatures.

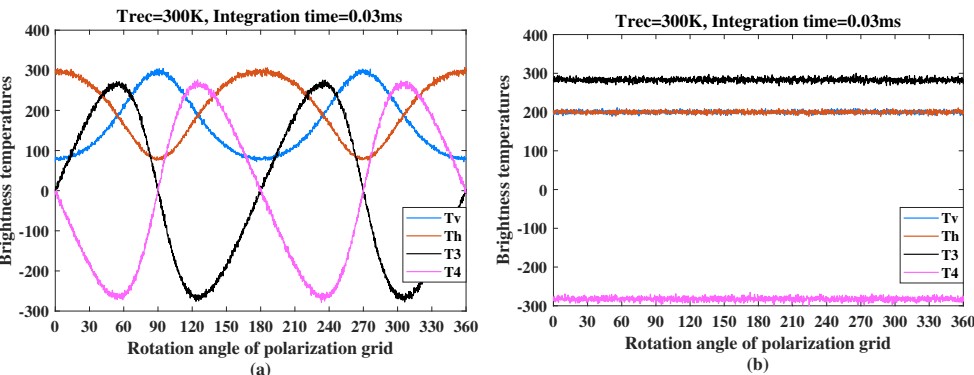

**Figure 8.** Four Stokes brightness temperatures. (**a**) CPCT signals; (**b**) ideal full-polarized signals.

*4.2. Validation*

4.2.1. Sensitivity

Sensitivity reflects the smallest resolvable brightness temperature at the input of the radiometer. The most commonly used sensitivity equation is:

$$\Delta T = \frac{T_{sys}}{\sqrt{B\tau}} = \frac{T_A + T_{rec}}{\sqrt{B\tau}},\tag{19}$$

where $T_{sys}$ is system noise, $T_A$ is antenna temperature, and $B$ is IF bandwidth. It can be seen that when the temperature of the antenna $T_A$ is determined, the sensitivity is mainly related to $T_{rec}$, $B$, and $\tau$. The square root of $B$ and $\tau$ are inversely proportional to the sensitivity, and $T_{sys}$ is directly proportional to the sensitivity. The bandwidth of this model is set to 750 MHz. Therefore, the simulation sensitivity is validated from different $T_{rec}$ and $\tau$.

First of all, $T_{rec}$ is set to 0 K, 200 K, 400 K, and 600 K, respectively; $\tau$ is 0.3 ms. After calibration, the output Stokes parameters in brightness temperatures are shown in Figure 9. It can be seen from Figure 9 that when $T_{rec}$ increases, the amplitudes of $V_3$ and $V_4$ of the CPCT do not change, but $V_v$ and $V_h$ increase accordingly. This is because $T_{rec}$ is used as the input to the vertically and horizontally polarized channels, so the amplitudes of $V_v$ and $V_h$ will increase accordingly. However, the components caused by $T_{rec}$ in $V_v$ and $V_h$ are irrelevant, so their correlation will not increase with the increase in $T_{rec}$.

The output Stokes parameters under different $\tau$ are also simulated. $\tau$ is set to 0.003 ms, 0.03 ms, 0.3 ms, and 3 ms, respectively, and $T_{rec}$ is set to 0 K. The simulation results are shown in Figure 10.

It can be seen that the amplitude of output voltages does not change as $\tau$ increases; however, the fluctuation component on the voltage will decrease significantly. Thus, $\tau$ has no effect on the amplitude of output voltage, but it will affect the sensitivity of the output result.

In order to calculate the sensitivity under different receiver noise and integration time, we set $B$ to 750 MHz, then calculated the actual equivalent bandwidth [24].

Taking channel $T_v$ as an example, the theoretical results and simulation results are shown in Figure 11. The straight lines indicate the theoretical sensitivity calculated by Equation (19), and the discrete dots indicate the simulated sensitivity calculated by the simulation model. It can be seen from that the theoretical and simulation values of the calculation results are consistent under different system noise and integration time, and the maximum error is less than 8%. This also verifies the reliability of the simulation model.

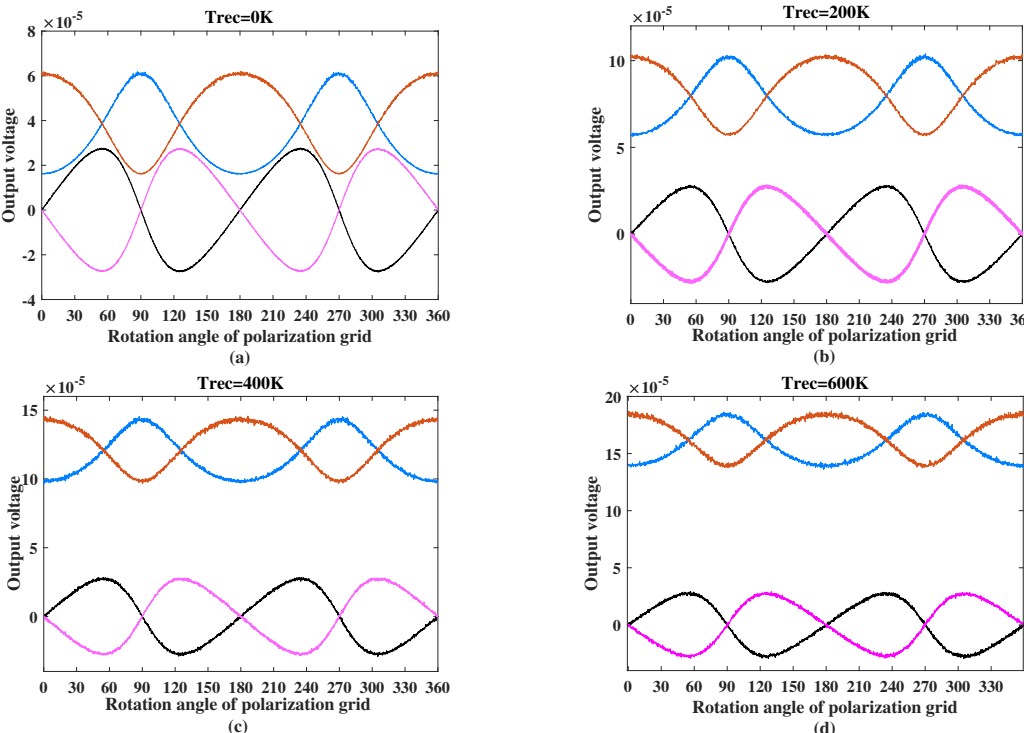

**Figure 9.** Output voltages under different $T_{sys}$ when $\tau$ is 0.03 ms. (**a**) $T_{sys}$ is 0 K; (**b**) $T_{sys}$ is 200 K; (**c**) $T_{sys}$ is 400 K; (**d**) $T_{sys}$ is 600 K.

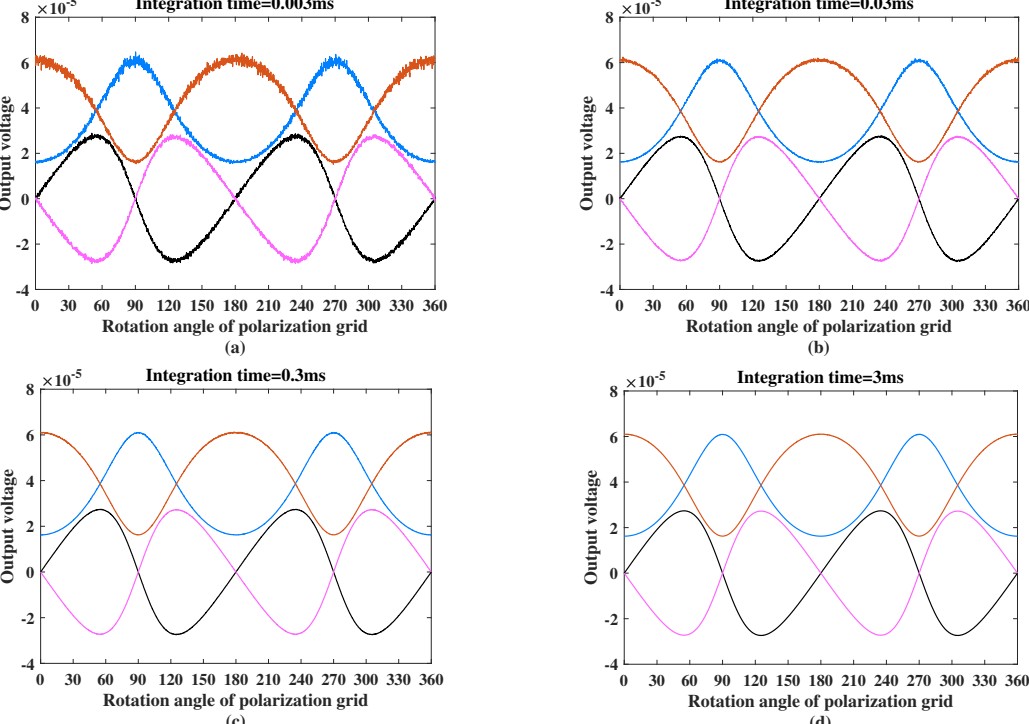

**Figure 10.** Output voltage under different $\tau$ when $T_{sys}$ is 0 K. (**a**) $\tau$ is 0.003 ms; (**b**) $\tau$ is 0.03 ms; (**c**) $\tau$ is 0.3 ms; (**d**) $\tau$ is 3 ms.

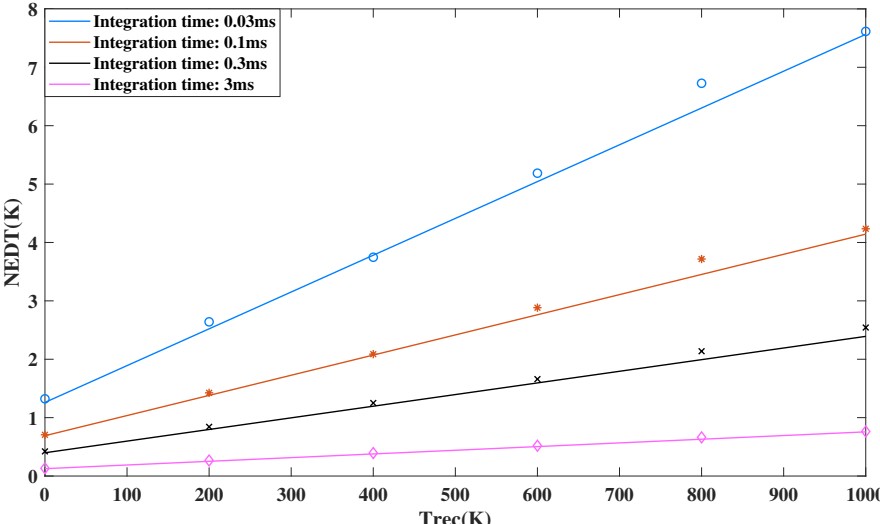

**Figure 11.** Theoretical and simulated sensitivity under different system noise and integration time.

### 4.2.2. Linearity

Good linearity can ensure the accuracy of output brightness temperature. When the gain is stable, linearity can be obtained by calculating the correlation coefficient between the theoretical brightness temperature and simulated brightness temperature. The theoretical brightness temperature of the system can be calculated by Equations (2) and (3), and the simulated brightness temperature can be calculated by the simulation model using the calibration matrix. The calculation formula of correlation coefficient is:

$$R = corr(T_M, T_B) = \frac{\sum_{i=1}^{N}\left(T_M^i - \bar{T}_M\right)\left(T_B^i - \bar{T}_B\right)}{\sqrt{\sum_{i=1}^{N}\left(T_M^i - \bar{T}_M\right)^2}\sqrt{\sum_{i=1}^{N}\left(T_B^i - \bar{T}_B\right)^2}}, \tag{20}$$

According to Equation (20), the linearity of the four channels is calculated. The linearity of $T_v$, $T_h$, $T_3$, and $T_4$ are greater than 0.9993, 0.9991, 0.9993, and 0.9997, respectively. This reflects the good linearity of the full-polarized radiometer simulation model.

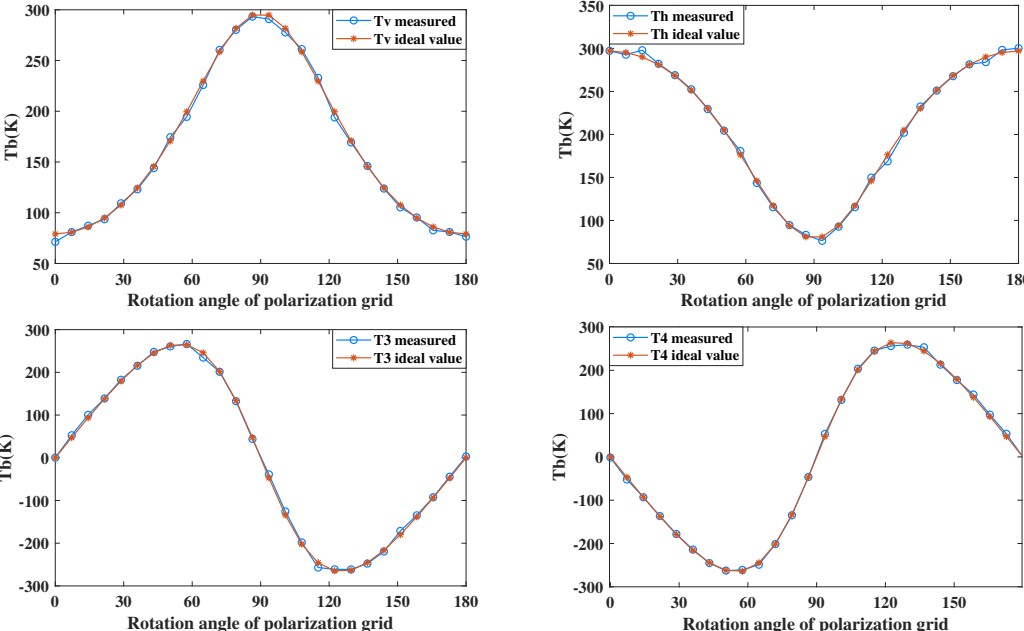

**Figure 12.** Linearity of four Stokes parameters calculated by simulation model.

Figure 12 shows a comparison between the theoretical and the simulated brightness temperature of a full-polarized radiometer when $T_{rec}$ is 300 K and $\tau$ is 0.03 ms. It can be seen that the simulation values are well consistent with the theoretical values in all four channels.

4.2.3. Calibration Residual Error

The calibration residual error $T_R$ is defined as the difference between the theoretical brightness temperature $T_T$ and the calculated brightness temperature $T_M$, which reflects the accuracy of the calibration result:

$$T_R = T_T - T_M. \tag{21}$$

Figure 13 shows the calibration residual errors of the four Stokes parameters of the simulation model with $T_{rec}$ of 300 K and $\tau$ of 3 ms. The horizontal axis represents the theoretical angle of the CPCT, and the vertical axis represents $T_R$.

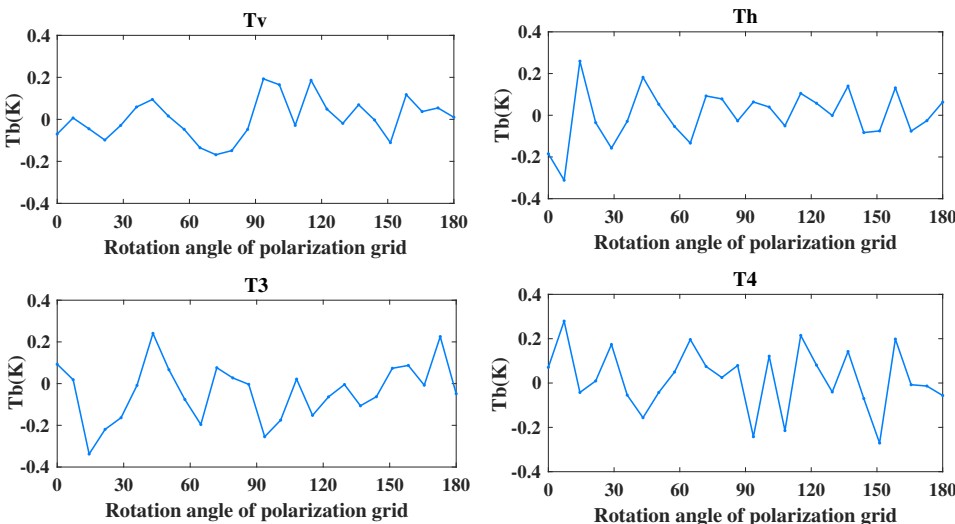

**Figure 13.** Calibration residual error of four Stokes parameters.

It can be seen from Figure 13 that the four Stokes parameters have a certain calibration residual error at each angle, which are all within $\pm0.34$ K. This error is mainly caused by measurement uncertainty and is within the range of sensitivity. The results also validate the accuracy of the simulation model and the correctness of the calibration process.

*4.3. Application Examples*

4.3.1. Analysis of Gain Imbalance

Gain imbalance is caused by the fact that the channel gain of the RF front-end is difficult to completely guarantee consistency, but whether it will affect the calibration result of the full-polarized signals needs further analysis. In this simulation, the gains of the four channels of the simulation mode are set to 70 dB, 71 dB, 72 dB, and 73 dB, respectively. Other simulation settings remain the same as the previous simulation.

Figure 14a shows the output voltages for the CPCT and hot and cold targets while $T_{rec}$ is 300 K and $\tau$ is 0.03 ms. It can be seen that due to the gain imbalance, the $V_v$ and $V_h$ of the hot target and cold target are not identical any more. $V_3$ and $V_4$ are zero, because $V_v$ and $V_h$ are still non-correlated. The four Stokes voltages of CPCT have also deviated. Although $V_v$ and $V_h$ of the CPCT are not identical, the maximum and minimum of their amplitude is still basically consistent with the $V_v$ and $V_h$ of the hot target and cold target. The $V_3$ and $V_4$ of the CPCT have also deviated as $V_v$ and $V_h$. The deviation is directly proportional to the gain imbalance. Through the results above, further calibration is performed. The

calibration process is the same as that described in Section 3. The calibration results are shown in Figure 14b.

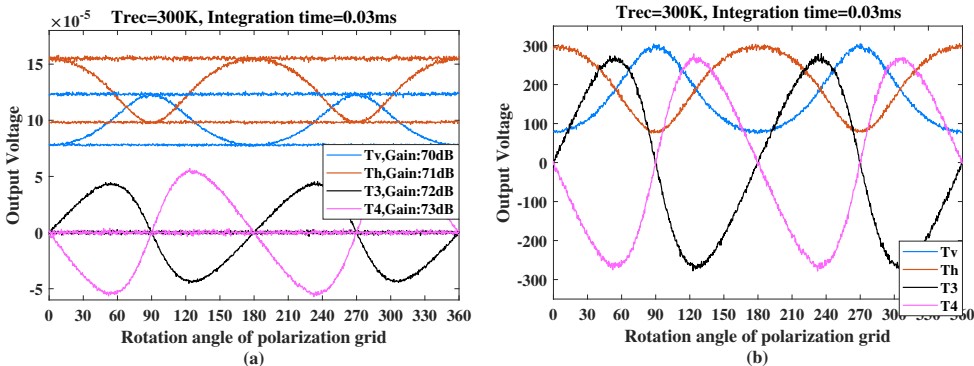

**Figure 14.** (**a**) Output voltages with imbalanced gain; (**b**) four Stokes parameters of CPCT after calibration.

It can be seen that the maximum and minimum of $T_v$ and $T_h$ after calibration are 305.17 K and 73.33 K, and the maximum and minimum of $T_3$ and $T_4$ are 279.36 K and −278.31 K, respectively, which are consistent with the output brightness temperature when the gain is equal, as was shown in Figure 8a. Therefore, the gain imbalance does not affect the accuracy of the output brightness temperature. Generally, the input power of ADC has a certain input range, and the signal can maintain good linearity within this range. Therefore, the gain imbalance only needs to make the input power meet the range of ADC.

### 4.3.2. Analysis of Quantization Error

The digital correlator quantizes the analog signals into digital signals. Quantization will inevitably lead to a decrease in measurement accuracy, so the quantization error is simulated and analyzed in this part.

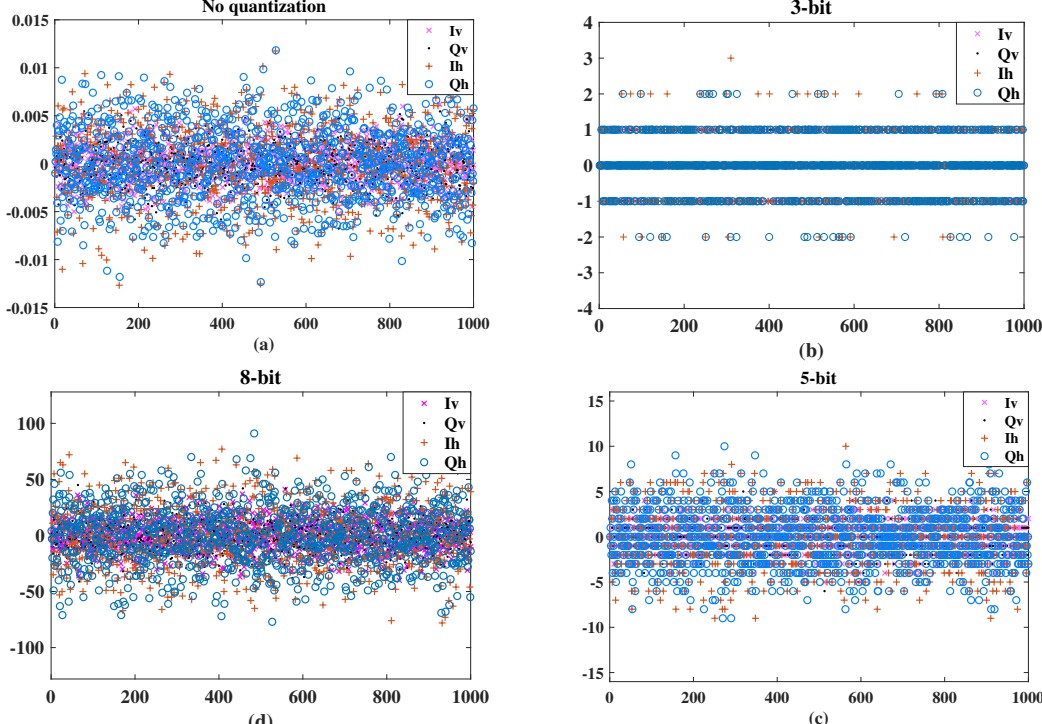

**Figure 15.** IQ signals after quantization. (**a**) No quantization; (**b**) 3-bit; (**c**) 5-bit; (**d**) 8-bit.

Figure 15 shows the quantized results of the IQ signals when using 3-bit and 8-bit quantization, respectively. As can be seen from the figure, when using 3-bit quantization, the amplitude of the IQ signals has only seven discrete values. When 8-bit quantization is used, the amplitude has 255 values, which is more consistent with the analog signals. Quantization mainly affects radiometer sensitivity. Therefore, we calculated the sensitivity of the simulation model at different quantization levels. In this simulation, the quantization level is 3-bit, 5-bit, and 8-bit.

The sensitivity results when $T_{rec}$ is 0 K and $\tau$ is 3 ms are listed in Table 3. In addition, it also gives the theoretical values of sensitivity without quantization. It can be seen from Table 3 that as the quantization level increases, the simulated sensitivity continues to decrease. At 3-bit, 5-bit, and 8-bit, the sensitivity error is 5.8%, 1.1%, and −1.3% of the analog sensitivity, respectively. In addition, by comparing the simulation sensitivity with the theoretical sensitivity, it can be seen that they are very close, which also verifies the full-polarized radiometer model.

**Table 3.** Comparison of sensitivity under different quantization levels.

| Quantization Level | Theoretical Value | Analog | 3-bit | 5-bit | 8-bit |
|---|---|---|---|---|---|
| Sensitivity (K) | 0.1261 | 0.1327 | 0.1404 | 0.1432 | 0.1309 |

## 5. Conclusions

This paper presented a simulation model of a full-polarized radiometer system. The simulation system consisted of input signal models and a fully polarized radiometer model. Input signals included CPCT signals, hot and cold calibration target signals, and ideal fully polarized signals. Compared with [25], it can directly generate a signal with phase information, which is more concise and clear. The input signal models were mainly based on WGN to generate noise signals for three targets. The fully polarized radiometer system model mainly included an RF front-end model and a digital back-end model. The most important part of the RF front-end model was the IQ demodulation module, which converted the input signals into IQ signals through two orthogonal LO signals. The digital back-end model mainly included quantization, multiple correlation and integration of the IQ signals, and, finally, it output the four Stokes voltages of the radiometer. The input and output relationship of CPCT signals and cold and hot calibration target signals were used to calculate the calibration matrix. Since there were four types of data and much more than five groups, by solving the overdetermined equations, the calibration matrix was calculated more accurately and conveniently as compared with [25]. In addition, we calculated the sensitivity, linearity, and calibration residual errors of the simulation model, and we verified the model by comparing them with the theoretical values. Finally, the simulation model was applied to the analysis of gain imbalance and quantization error. The results showed that as long as the gain imbalance met the input range requirements of the ADC, it did not affect the output brightness temperature. At 3-bit, 5-bit, and 8-bit, the simulation sensitivity error was 5.8%, 1.1%, and −1.3% of the analog sensitivity, respectively. In general, the simulation model in this paper has high simulation accuracy and can be applied to the error analysis of fully polarized digital correlation microwave radiometers, which will have very important significance. However, due to time limits, the maximum integration time of this model was set to 3 ms, and the amount of data was insufficient, which led to the situation of the sensitivity of the 8-bit quantization being less than that of the analog result. Due to lack of space, the effects of the antenna pattern, antenna attitude, atmospheric channel response, and so on were not discussed in this paper. The errors caused by the quantization, the phase delay plate, and other factors also need to be analyzed in future work. With the development of a full-polarized radiometer, we will input a series of real parameters into the simulation model to make the digital model, and we expect that the digital twin simulation system model will work like a real full-polarized radiometer.

**Author Contributions:** J.D. performed the simulation and analyzed the results; H.L. provided the parameter data of the cpct calibration source; J.D. wrote this paper; Y.D. guided this paper; Z.W. and X.T. edited the article. All authors have read and agreed to the published version of the manuscript.

**Funding:** This work was supported by the National Natural Science Foundation of China (Grant Nos. 41771405). http://www.izaiwen.cn/, accessed on 29 November 2021.

**Institutional Review Board Statement:** Not applicable.

**Informed Consent Statement:** Not applicable.

**Data Availability Statement:** Not applicable.

**Acknowledgments:** The authors would like to thank the Key Laboratory of Microwave Remote Sensing for providing the actual parameters of the receiver and CPCT system.

**Conflicts of Interest:** The authors declare no conflict of interest. The funders had no role in the design of the study, nor in the collection, analyses, or interpretation of data, as well as no influence in the writing of the manuscript, nor in the decision to publish the results.

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
