# Peer review of "A Digital-Simulation Model for a Full-Polarized Microwave Radiometer System and Its Calibration"

_remotesensing, doi:10.3390/rs13234888_

Round 1

Reviewer 1 Report

The manuscript describes a calibration target and a digital simulation model for a full-polarization radiometer. Some examples of applications of the simulator to analyze effects of radiometer gain imbalance and quantization error are given. The manuscript is well-organized and its development is generally logical, but the points discussed below should be addressed.

1. Eq.1 defines the four Stokes parameters used in this paper. In the usual definition, the third Stokes parameter is related to linear polarization of the signal and the fourth Stokes parameter measures circular polarization (as in eqs.2 and 3 in ref.22). The authors do not make these identifications, but it would be helpful to readers to indicate whether the usual definitions apply. I am concerned about V3 in particular, because  by the usual definition it should be
V3= <Iv Ih + Qv Qh>.

2. On Figs. 6-8 the vertical scale is labeled "Output voltage" but the values appear to be brightness temperatures.

3. Section 4.2.2 uses the correlation coefficient between the simulated and theoretical brightness temperatures as a measure of linearity. In a real radiometer, nonlinearity could be introduced by the amplifiers, but eq.15 simply uses the scalar constant A as gain. Therefore, the simulator does not model a real radiometer in this respect.

Author Response

In tex File ,line 6 there is One line of code:

“\usepackage{changes}”,(line 6)  means it will show the track changes in the PDF file,if  you want the final version directly shown in PDF,just change the code into “\usepackage[final]{changes}” . If you can't run the.tex File (.tex) , there are also two PDF files named track changes and final version.

1.The type of correlator we use and basic principleshas been described, and The misexpression of V_3   has been corrected.

2.Change the vertical scale label from "Output voltage" to " brightness temperatures "of figure 6-8 is modified.

3.The longer paragraphs have been adjusted in 2.1/3.1.3/4.3.1/4.3.2 / 5-conclusion. Split the long paragraph into several small paragraphs.

4 In order not to cause misunderstanding, ‘measured brightness temperatures’has been changed into ‘calculated brightness temperature’ .

Reviewer 2 Report

The authors of the manuscript did a decent job describing their work, and what they did is very interesting. I have the following comments: 

1) General: Paragraphs are long and contain multiple ideas. Please make paragraphs smaller and let them address specific ideas. 

2) It was not clear to me whether the authors are trying to compare the results of the simulation model they built to an actual hardware representing fully polarized radiometer or what? So what do you mean by "measured brightness temperatures"? 

3) Is it possible to compare results from this model to brightness temperature from actual radiometer so you can obtain real measurements? 

Author Response

(The authors gave the same response as above.)
